# Globular pattern formation of hierarchical ceria nanoarchitectures
Noboru Aoyagi [1] ✉, Ryuhei Motokawa [2], Masahiko Okumura [3], Yuki Ueda [2], Takumi Saito [1,4], Shotaro Nishitsuji [5], Tomitsugu Taguchi[6], Takumi Yomogida [7], Gen Sazaki [8] & Atsushi Ikeda-Ohno [1]

Dissipative structures often appear as an unstable counterpart of ordered structures owing to fluctuations that do not form a homogeneous phase. Even a multiphase mixture may simultaneously undergo one chemical reaction near equilibrium and another one that is far from equilibrium. Here, we observed in real time crystal seed formation and simultaneous nanocrystal aggregation proceeding from $Ce^{IV}$ complexes to $CeO_2$ nanoparticles in an acidic aqueous solution, and investigated the resultant hierarchical nanoarchitecture. The formed particles exhibited two very different size ranges, resulting in further pattern formation with opalescence. The hierarchically assembled structures in solutions were $CeO_2$ colloids, viz. primary core clusters (1–3 nm) of crystalline ceria and secondary clusters (20–30 nm) assembled through surface ions. Such self-assembly is widespread in multi-component complex fluids, paradoxically moderating hierarchical reactions. Stability and instability are not only critical but also complementary for co-optimisation around the nearby free energy landscape prior to bifurcation.

Complex fluids with multiple components have attracted considerable attention over the last quarter of a century, after the serendipitous discovery of their phase separation within biological cells and biomimic systems[1–3]. For in vivo or in vitro molecular separations, concentration control is crucial[4]. Meanwhile, inorganic polymers exist in various phases, such as glasses, liquid crystals, and colloids[5–9]. Among these, colloidal particles inliquid media like sol-gels have been of interest in engineering and fundamental science, including soft condensed matter physics. However, comprehensive insight into multi-component mixtures with strong or weak interactions remains challenging, particularly considering the balance of colloidal particles dispersed and surrounded by labile ligands (e.g., water, acid, or organic solvent) undergoing rapid exchange reactions.

Water plays a special role in forming amorphous suspensions or precipitates of metal oxide ions due to their rapid hydrolytic reactions. A closer examination of these materials reveals that metal aquo complexes form direct bonds via hydroxo-bridging[10]. At equilibrium, deprotonation occurs within the complex. The corresponding species are linked in successive equilibria to generate– (hydr)oxidemetal clusters due to the remarkable attraction among tetravalent heavy metal ions ($Ce^{4+}$, $Th^{4+}$, $U^{4+}$, $Np^{4+}$), even at low pH[11–13]. The generated hydroxide-bearing polymers often display disordered structures and amorphous phases in the liquid state, and the crystallites grow from sub-micrometre sizes. With time, the particle size increases by growth and agglomeration, and the particle size and structure in turn influence the surface area of the nanostructured solid-state material. The solid phase is negatively charged due to deprotonation, and thus, the surface moiety is surrounded by counter-cations. The resulting interparticle repulsion mainly stems from the electrostatic force that competes with the attractive van der Waals force[14]. Understanding the ion–surface interaction within these systems is critical, because these two opposing forces (attraction by hydroxides and repulsion by cations) are key to the design and control of nanoparticle architecture in mesoscale engineering. The standard theory of colloidal dispersions, on the other hand, is not historically simple [15].

Here, we consider a representative system consisting of $Ce^{4+}$ aquo ions, $Ce^{IV}$ complexes, and $Ce^{IV}$ nanoparticles to reveal the intrinsically multiscale, hierarchical interaction when such nanostructures are formed. Ceric ammonium nitrate (CAN) displays unique chemistry in organic solvents and water[16,17]. As a strong oxidising agent, it is versatile in catalytic syntheses with acetonitrile, methanol, or acetone as the typical media. The reaction

[1]Advanced Science Research Centre (ASRC), Japan Atomic Energy Agency (JAEA), Tokai-mura Ibaraki 319-1195, Japan. [2]Materials Sciences Research Centre (MSRC), Japan Atomic Energy Agency (JAEA), Tokai-mura Ibaraki 319-1195, Japan. [3]Centre for Computational Science and e-Systems, JAEA, Kashiwa Chiba 277-0871, Japan. [4]Nuclear Professional School, School of Engineering, The University of Tokyo, Ibaraki 319-1188, Japan. [5]Graduate School of Science and Engineering, Yamagata University, Yonezawa, Yamagata 992-8510, Japan. [6]National Institutes for Quantum Science and Technology, Takasaki-shi, Gunma 370-1292, Japan. [7]Nuclear Science and Engineering Centre (NSEC), Japan Atomic Energy Agency (JAEA), Tokai-mura Ibaraki 319-1195, Japan. [8]Institute of Low Temperature Science, Hokkaido University, Kita-ku Sapporo 060-0819, Japan. ✉e-mail: aoyagi.noboru@jaea.go.jp

requires less CAN than the equivalent amount of reactants and therefore does not typically result in concentrated $Ce^{4+}$ in the solution. Demars et al. reported that a $Ce^{IV}$ oxo-bridged dinuclear complex, viz. $[Ce^{IV}–O–Ce^{IV}]^{6+}$, is critical in catalysing C–O bond formation by accepting electrons on two adjacent centres within the same complex[18]. Relatively simple geometries have been reported for CAN in the organic phase, whereas its geometries in the aqueous phase are far more complex due to multiple hydrolytic reactions, even at very acidic or under dilute conditions[19,20]. The vast majority of experimental studies on CAN in water employ bottom-up approaches, starting from the monomer, dimer, trimer, tetramer, hexamer, *etc.* Within these complexes, the hydroxyl group ($OH^-$) provides strong linkages between multiple $Ce^{4+}$ in various ways. Solid $CeO_2$ also contains this robust framework with deprotonated $OH^-$ groups. Ikeda-Ohno et al. distinguished double oxo-bridged dimers and large crystals of $CeO_2$ using extended X-ray absorption fine structure (EXAFS) analysis[20]. However, reports of atomically precise $CeO_2$ nanocrystal synthesis remain limited[21]. Although researchers continue to fill gaps in the cluster size from monomer, dimer, and hexamer to $\{Ce_{24}^{IV}\}$, $\{Ce_{38}^{IV}\}$, $\{Ce_{40}^{IV}\}$, and very recently $\{Ce_{100}^{IV}\}$, refs. [19–25]; that is, the results remain incomplete. Another future research topic is controlling the aggregation of $Ce^{IV}$ nanoclusters. CAN possesses two moles of anion relative to one mole of cation, presenting a typical multiscale system with charged ions or complexes, nanocrystal growth, and interactions among colloidal particles (Fig. 1). Liu et al. have unveiled the versatile polyoxometalate macro ions using the transition metals to form vesicles, cavities and spheres[26,27]. Remarkably, Smarsly et al. reported the first observation of hierarchical growth and aggregation of ceria colloid[28]. Regardless of these successful efforts to obtain clusters, there has been no other hierarchical patterns except globular aggregation. The lack of knowledge on how to control the alignment of aggregated particles based on nanoscale modelling for globular colloid with tetravalent metal ion is an unparalleled approach to gain an in-depth understanding of the patterns formed by these materials, which may exhibit photophysical properties - opalescence or iridescence[29]. This study aims to observe in real time the crystal seed formation, aggregation and further extensive flocculation toward $CeO_2$ nanoparticles and subsequent in-situ patterns in a CAN-$HNO_3$ solution and carry out the corresponding modelling. The system involves two/three spatiotemporally different patterns commonly emerge in reaction-diffusion processes of condensed complexed fluids with multiple components. Thus, it is relevant to a broad range of phenomena in the design and synthesis of metal oxides with precisely controlled size, structure and texture. Furthermore, the resulting nanostructured $CeO_2$ has potential applications in catalysis and other areas wherein the interface of nanoarchitectures is critical.

## Results

### Transmission electron microscopy (TEM)

Molecular-scale imaging provides in-depth data regarding the crystal growth and concomitant phase transition[30]. Figures 2 and 3 show TEM images of dried CAN solutions at different concentrations in 0.1 M $HNO_3$. Cases with the higher total Ce concentration ($C_{Ce^{IV}} = 0.50$ M) are shown in Fig. 2a–e, while Fig. 2f, g display the Fourier transforms of Fig. 2c and the single domain of a primary cluster in Fig. 3c, respectively. The primary clusters are homogeneously dispersed at higher $C_{Ce^{IV}}$ with an average diameter of 1–3 nm (Fig. 2a), and they exhibit single 1nanocrystalline structures with diffraction patterns consistent with the space group $Fm\bar{3}m$ (Fig. 2f, g). Moreover, the lattice parameters $a = b = c = 0.541$ nm and $\alpha = \beta = \gamma = 90°$ are common in numerous $CeO_2$ polycrystals[31].

These primary clusters do not appear to aggregate randomly, but rather exhibit a linear structure along the edge of the holey-carbon layer in the copper grid. Ball and Witten proposed the primitive Sutherland's ghost model to distinguish two types of irreversible aggregation: the accretion of individual particles and the successive aggregation of clusters comparable in size[32]. The drying process used for sample preparation was too rapid to deform the interaction among these clusters and the substructure in the primary core, hence the irreversible aggregation due to the lack of the outer solvent in the diffusive medium. This step may lead to the successive aggregation with particle size remains almost at constant[33,34]. However, there could be some scopes to revisit a more detailed discussion of the interaction between a larger aggregated particle and a smaller primary particle, which seems interesting but is beyond the scope in the present study[15].

Conversely, at the lower concentration of $C_{Ce^{IV}} = 50.0$ mM in 0.1 M $HNO_3$, the TEM images reveal a large architecture with 20–30 nm colloids (Fig. 3a–c) forming an aggregate-flocculate of loosely bound secondary particles (cf. Supplementary Note 1 for conditions). These colloids appear spherically or hexagonally filled (Fig. 3c) with sustained polycrystalline structures (Fig. 3d for the single domain of a secondary particle in 3b). The genesis of the secondary clusters is critical, because the TEM images only reveal these two nanoparticle types that are very different in size. A closer examination of the colloidal sphere shows that the large particle is composed of assembled primary particles (Fig. 2g), indicating strong cluster–cluster interactions in the liquid state, whereas the sample's nature changes during drying.

The same drying step was used at both $C_{Ce^{IV}}$ concentrations. The cluster–cluster aggregation is reversible because the spatial distribution is random (Fig. 2b), instead of the successive formation of primary cluster aggregates. Therefore, different interactions exist within the primary and secondary clusters. The interaction between the charged primary clusters is due to the surface complexation by hydroxo-bridging or anion coordination. We attempted to measure the zeta potential, in addition to the effective interparticle potential in the strong acid. However, the measurement failed because the 0.1 M $HNO_3$ corroded the copper electrode. Conversely, the secondary particles are sufficiently bulky and therefore we could ignore their electrostatic interactions. The cluster-cluster interaction, which led to colloidal growth from monodispersed nanocrystals to inter-cluster aggregates, was considered (cf. Supplementary Note 2)[35]. For the ideal colloidal interaction, Derjaguin-Landau-Verwey-Overbeek (DLVO) theory[14,36] is based on the equilibrium of forces, such as the electrostatic force of the double layer (dispersion repulsion) and the van der Waals forces (aggregation attraction), which is not valid due to the rather short distance. While this $CeO_2$ system at concentrated liquid, c.a. $C_{Ce^{IV}}$ becomes more than ~0.5 M, followed the classical theory, non-DLVO interactions possibly predominates in hydroxo-bridging surface species, which is a type of structure effect specific to the eigen shapes (non-spherical) of nanocrystals (such as primary clusters involving hundreds of Ce atoms). In particular, the DLVO theory collapses in the phase change from primary clusters to secondary clusters— under the dilute $C_{Ce^{IV}}$ condition at ~0.05 M. A more comprehensive

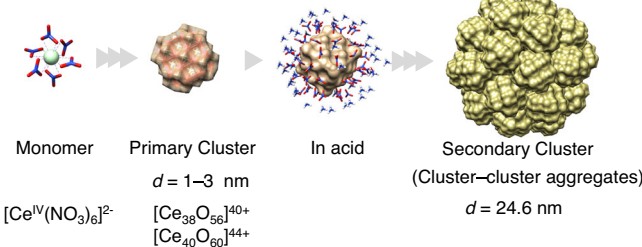

| Monomer | Primary Cluster | In acid | Secondary Cluster (Cluster–cluster aggregates) |
|---|---|---|---|
| | $d = 1$–3 nm | | $d = 24.6$ nm |
| $[Ce^{IV}(NO_3)_6]^{2-}$ | $[Ce_{38}O_{56}]^{40+}$ $[Ce_{40}O_{60}]^{44+}$ | | |

**Fig. 1 | Two-step model for $CeO_2$ crystal growth and subsequent cluster aggregation, starting from the surface of the nitrate complex in a complex fluid (left).** Oligomerisation via hydroxo-bridging has been reported[20]. The number of cores within this primary cluster is undetermined. In concentrated $HNO_3$, nitrate anions coordinate to the primary cluster surface. Moreover, the ammonium cations surround the nitrate anion shell (shown in the third panel "In acid"), resulting in electrical triple layers (water is omitted for clarity, and ammonium ions are shown for representational purposes). The secondary cluster is an aggregate of primary clusters at the local potential minimum (contributed by the electrostatic and van der Waals potentials) in terms of the classical Derjaguin-Landau-Verwey-Overbeek theory. The atom colours are as follows: Ce (pale blue in a monomer and yellow in a cluster), N (blue), O (red), and H (white). Each surface depicts the monomer of the primary cluster.

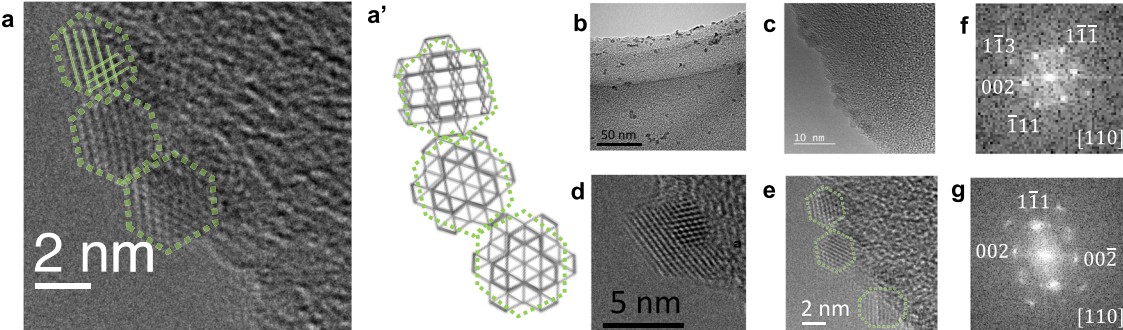

**Fig. 2 | High-resolution transmission electron microscopy (HRTEM) images of CeO₂-containing "primary" nanocrystals on the carbon-coated copper microgrid.** The total concentration of added ceric ammonium nitrate is $C_{Ce^{IV}} \approx 0.50$ M in 0.10 M HNO₃. **a** HRTEM image at the highest resolution. **a'** Schematic drawing of the aggregate-flocculate (Sutherland's ghost model). **b, c** HRTEM images at different resolutions. **d** A twin crystal. **e** TEM image displaying different aggregate-flocculate from that shown in (**a**). **f, g** Selected area electron diffraction patterns indicating particle crystallinity. Scale bars: (**a**) 2 nm, (**b**) 50 nm, (**c**) 10 nm, (**d**) 5 nm, (**e**) 2 nm, (**f**) 10 nm⁻¹, and (**g**) 10 nm⁻¹.

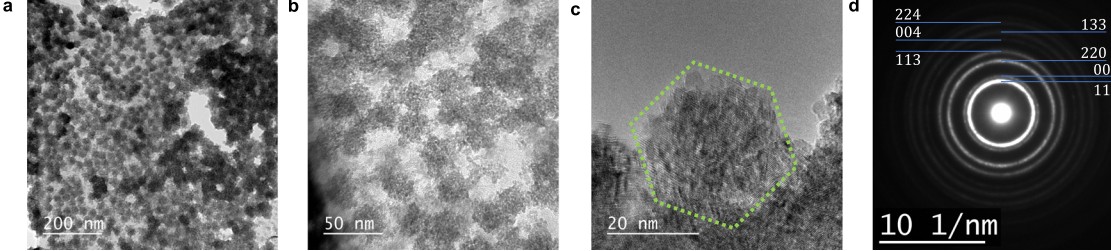

**Fig. 3 | High-resolution transmission electron microscopy (HRTEM) images of CeO₂-containing "secondary" nano architectures. a–c** Low-magnification TEM images of aggregated "secondary" clusters on the carbon-coated copper microgrid. The total concentration of added ceric ammonium nitrate was $C_{Ce^{IV}} = 50.0$ mM in 0.1 M HNO₃. **d** Selected area electron diffraction pattern indicating particle polycrystallinity (the Debye-Scherrer ring). Scale bars: (**a**) 200 nm, (**b**) 50 nm, (**c**) 20 nm, and (**d**) 10 nm⁻¹.

approach in understanding the long-range interaction is necessary. The Sogami-Ise theory with pair $G$ potential gives an attractive tail to the detailed discussion[37–39]. As far as we understand the universality with difficulty, the other attraction—also an electric interaction generated by counterion condensation—could be considerably larger than the van der Waals force. We adopt the interpretation of surface interaction based on Sogami-Ise theory and its relevant theory to have a better picture of the hierarchical architectures.

## Raman spectroscopy

Figure 4 shows the Raman spectra of pristine CAN and CeO₂, in addition to CAN dissolved in 0.1 M HNO₃ at three different concentrations ($C_{Ce^{IV}} = 1.00$, 0.50, and 0.05 M). All spectral peaks for pristine solid-state CAN (spectrum **a** in Fig. 4) are assigned to well-known modes corresponding to the vibrations of NO₃⁻ and Ce^{IV}–O–Ce^{IV} (the "–O–" is from a bridging nitrate). The intense band at 252 cm⁻¹ is assigned to the Ce–O–Ce vibration, and the weak bands at 750 and 1035 cm⁻¹ are assigned to nitrate bound to Ce⁴⁺[18]. The most intense band in the spectrum of CAN at 457 cm⁻¹, which is located close to the $F_{2g}$ ($T_{2g}$ symmetry) triply degenerate Raman-active mode of CeO₂ powder (466 cm⁻¹), does not correspond to the solid-state pristine CAN or CAN-HNO₃ solutions (Fig. 4b, c, e, f)[40–42]. Instead, this peak (457 cm⁻¹) corresponds to the $F_{2g}$ mode. Considering the resolution under the experimental condition (~4 cm⁻¹), the two peaks at 457 and 466 cm⁻¹ are certainly different with a non-negligible phonon energy of 9 cm⁻¹. The bands at approximately 252 and 610 cm⁻¹ in the spectra of CAN powder and CAN solution at higher concentrations (spectrum **b** in Fig. 4) indicate non-negligible amounts of oxygen vacancies compared to those within CeO₂ powder[43–45]. These trends are especially remarkable in the batches containing the primary clusters. Conversely, at the relatively high concentrations of 1.00 and 0.50 M, the Raman signals are more distinct than those in the spectra of the secondary clusters. The peak at 606 cm⁻¹ on

spectrum **d** in Fig. 4 indicates the crystalline structure of the primary clusters, and the scattering tensors of the band are $A_{1g}$, $E_g$, and $F_{2g}$[41]. This band was observed for all samples, although its intensity decreased at higher concentrations (labelled with "*" on spectra **b** and **c** in Fig. 4). Consistent with TEM (c.f. Fig. 2b), these batches contain the primary clusters as a significant component. In the supplementary information we describe further details from Raman spectroscopy. Briefly, the result is as follows:

$$\Gamma = a + (b/D) \tag{1}$$

where $\Gamma$ (cm⁻¹) is the half-width at half-maximum of the observed Raman line, and $D$ (nm) is the particle radius (*see* Supplementary Note 3).

## SAXS

SAXS is very informative, particularly for analysing the average physical properties of colloidal particles. Figures 5a, b show the observed SAXS scattering intensity $I(q)$ for a solution containing $C_{Ce^{IV}} = 50$ mM in 1.0 M HNO₃. Here, $\lambda$ and $2\theta$ are the wavelength of the incident X-ray and the scattering angle, respectively, and $q = (4\pi/\lambda) \sin \theta$ is the magnitude of the scattering vector. Figure 5a displays the time evolution of SAXS patterns for the solution maintained at pH 0.771 for $t_0 = 0$ d (black circles), $t_1 = 3$ d (red circles), $t_2 = 4$ d (green circles), and $t_3 = 5$ d (blue circles) after sample preparation. The SAXS pattern at 0 d shows weak scattering intensity over a wide $q$ range, indicating the absence of a nanoscale ordered structure in the solution. In sharp contrast, the patterns at ≥3 d indicate that an ordered structure appears spontaneously in the solution and grows with time. Specifically, the scattering intensity at low $q$ ($q < 1.0$ nm⁻¹) gradually increases with time, whereas that at high $q$ ($q > 1.0$ nm⁻¹) barely changes. There is no significant change at $t > 5$ d. These SAXS patterns come from two contributions: (i) $I_p(q)$ at $q > 1.0$ nm⁻¹ due to the primary clusters of the m-mer {Ce_m^{IV}}, and (ii) $I_s$ at $q < 1.0$ nm⁻¹ due to the aggregates (secondary

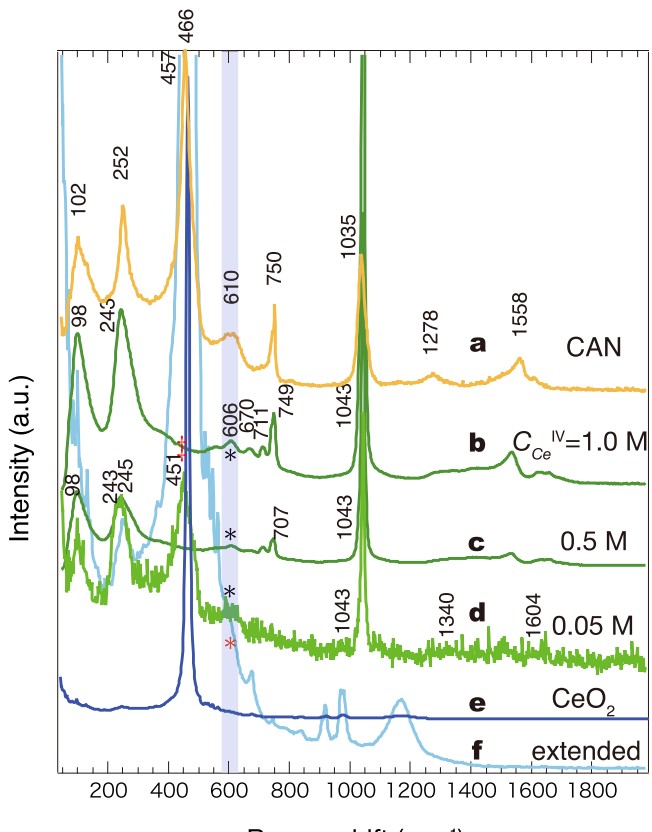

**Fig. 4 | Raman spectra of CeO₂-containing powder materials and CeO₂-containing primary nanocrystals in HNO₃. a** Ceric ammonium nitrate (CAN) powder dried at 100 °C for 3 h. CAN in 0.1 M HNO₃ at (**b**) $C_{Ce^{IV}}$ = 1.0 M, (**c**) $C_{Ce^{IV}}$ = 0.5 M, and (**d**) $C_{Ce^{IV}}$ = 0.05 M. **e** As-received CeO₂ powder. **f** Spectrum of CeO₂ powder magnified in the y-direction to highlight the very weak peaks.

clusters). Therefore, the mean size of the secondary clusters likely increases with time, whereas that of small clusters remains almost constant. The SAXS patterns are reproduced by summing $I_p$ and $I_s$, which assume the form factor of spherical particles with the respective radii of $R_p$ and $R_s$:

$$I(q) = A \left[ \frac{3\left\{ \sin\left(qR_p\right) - \left(qR_p\right)\cos\left(qR_p\right) \right\}}{\left(qR_p\right)^3} \right]^2 \\ + B \left[ \frac{3\left\{ \sin\left(qR_s\right) - \left(qR_s\right)\cos\left(qR_s\right) \right\}}{\left(qR_s\right)^3} \right]^2 \quad (2)$$

where $A$ and $B$ are proportionality constants. The $R_s$ distribution is considered to precisely reproduce the SAXS intensity distribution. Thus, we use a Gaussian distribution with standard deviation $\sigma_R$, while the $R_p$ distribution is ignored because of the invisibility of the region of $q > \frac{1}{R_p}$ in Fig. 5a. The solid lines in Fig. 5a represent the best-fit theoretical scattering curves using Eq. (2) with the refined parameters listed in Table 1. $R_p$ is constant at approximately 0.8 nm regardless of $t$, whereas $R_s$ increases from 9.2 nm at $t_0$ = 3 d to 12.4 nm at $t_1$ = 5 d. Remarkably, the first peak of $I_s\left(q\right)$ observed at $q \approx 0.5$ nm becomes sharper with $t$. Moreover, $\frac{\sigma_R}{R_s}$ changes from 0.14 (3 d) to 0.10 (5 d), indicating transformation of the particle shape to a more spherical (or highly symmetrical) structure due to spatial rearrangement of the primary clusters within the aggregates. Hence, colloidal aggregation proceeds while maintaining the primary cluster size. Additionally, the crystal growth of primary particles and cluster-cluster aggregation are at non-equilibrium during the observation period of several hours [19].

## Discussion

Our main aim was to understand the two hierarchically different intra/interparticle interactions governing the particle shapes. The primary cluster grows as a single crystalline phase following initial seed nucleation in the acidic solution. The first concern is whether the CeO₂ nanocrystal grows through the connection of building blocks based on Ce–(OH)–Ce bonding, including subsequent deprotonation to form the solid oxides. This hypothesis may lead to "proton ambiguity"—the difficulty of establishing the origin of a released proton experimentally[46]. Hence, an approach combining spectroscopic techniques (EXAFS and Raman spectroscopy) with theoretical calculations such as density functional theory (DFT) and molecular dynamic simulation is appropriate. Ikeda-Ohno et al. confirmed dinuclear complex formation using EXAFS analyses and DFT calculations[20]. The protons on the double hydroxo-bridge between two Ce ions are predominantly present within $[Ce^{IV}_2(\mu_2\text{-}OH)_2(H_2O)_{14}]^{6+}$[20]. The monomer-to-dimer complex formation is the nucleation step, followed by reaction limited aggregation (RLA) that could continuously build up the molecular blocks and form nano primary clusters (Supplementary Fig. S1c). Concerning clusters, Mitchel et al. compared a distorted CeO₂ nanocrystal structure to the bulk and found a slight disorder in the boundary lattice structure[21]. Their finding suggests that excess protons or defects may exist within the crystal structure of pure CeO₂. The characteristic Raman band at 451 cm⁻¹ is common for condensed ensembles and the sample with $C_{Ce^{IV}}$ = 0.05 M, but the assignment of this band remains unclear (Fig. 4d) because it was not observed for the samples with $C_{Ce^{IV}}$ = 1.0 and 0.5 M. Conversely, the band also appears very specific to a chemical bond formed within the coarse-grained clusters. One plausible assignment is a phonon band associated with the monodispersed colloidal particles. Herein, the Raman spectrum reveals a mean particle size of 0.98 nm, which is roughly comparable to 1.68 nm (Table 2). Based on these findings, the average number of Ce atoms in the primary cluster is $\bar{m} \sim 10^2$. Reported chemical species in the literature range from the monomer complex to a 40-mer as one of the metastable groups[18,19,21], in which the crystal growth almost stops at a certain level of reaction due to structural optimisation across the local free energy landscape. This is essential for understanding primary clusters that appear by chemical reaction. (The fraction for each particle size is given in Supplementary Fig. S1). Therefore, 24-, 38-, and 40-mers are not magic numbers by the assumption of a continuous distribution of $\bar{m}$; additionally, single crystals with a slightly larger $\bar{m}$ could be isolated because they are more stable and form a vast majority in complex fluids. The other case generating the meso-crystals is reported by Soroka et al.[25]. The radiation doses provide necessary steps for the aggregation of Ce^IV ions produced by γ-ray from Ce^III ions because there is Ce^IV lacking in the initial condition in their experiment. Interestingly, no matter how much the total dose induced within their investigation, the hierarchical diameters resulted in two ranges: 1–3 nm and 15–30 nm which is identical to our result. This is not a coincidence but the possible to fill the two potential minima of the Sogami-Ise theory[15,39] that contains dispersed macromolecules in the presence of the dilute ions. In terms of RLA, it is significant to extend the chemical reaction model to include the self-assembly of primary clusters. The nucleation and crystal growth of CeO₂ nanoparticles in HNO₃ solution was observed at different solution pH. When the orange-coloured CAN powder dissociates in an acidic solution, Ce⁴⁺ forms various complexes with NO₃⁻, H₂O, and OH⁻ generated by water dissociation. During seed crystal nucleation, the bridging ligand is significant because direct Ce^IV–Ce^IV interaction is unfavourable in aqueous solutions. The hydrolysis may lead to the corresponding nucleation, and the nitrate anion is unlikely to bridge donors except in the solid state. A higher pH drives the following reaction equilibrium to the right:

$$m\text{Ce}^{4+} + n\text{OH}^- \rightleftharpoons \text{Ce}_m(\text{OH})_n^{(4m-n)+} \quad (3)$$

where $m$ = 1 and $n$ = 0–4. Details about our modelling is described in the Methods section as *self-assembly*. Now, we discuss the secondary clusters. We compare two models to describe the aggregation and dispersion

Fig. 5 | Double-logarithmic plots of the small-angle X-ray scattering (SAXS) patterns and the contrast-matching small-angle neutron scattering (CM-SANS) patterns ($I_{obs}(q)$ vs. q). a Profiles as a function of time for $C_{Ce^{IV}}$ = 50.0 mM in 1.0 M HNO₃. The solid line is the best-fit theoretical profile obtained using Eq. (2) combined with the characteristic parameters listed in Table 1. b SAXS profiles at different pH levels adjusted using NaOH (50 wt.% ≈ 19.7 M). The line colours (pH values) are: magenta (5.59), purple (1.66), green (1.05, 1.070), yellow (0.90–0.82), and black (0.77). The yellow lines were measured at 20-min intervals. c Double-logarithmic plots of the CM-SANS profile for the solution with $C_{Ce^{IV}}$ = 50.0 mM in 1.0 M HNO₃ using a solvent, D₂O/H₂O mixture (= 0.6618/0.3382, v/v), where the SLD of D₂O/H₂O mixture and CeO₂ crystals agree with each other in this condition. Inset shows a Guinier plot, where the plot of ln$I$(q) versus $q^2$ is linear for $qR_{g,am}$ < 1, enabling $R_{g,am}$ to be estimated from the slope. The black solid line in the inset shows the theoretical best fit by the Guinier approach.

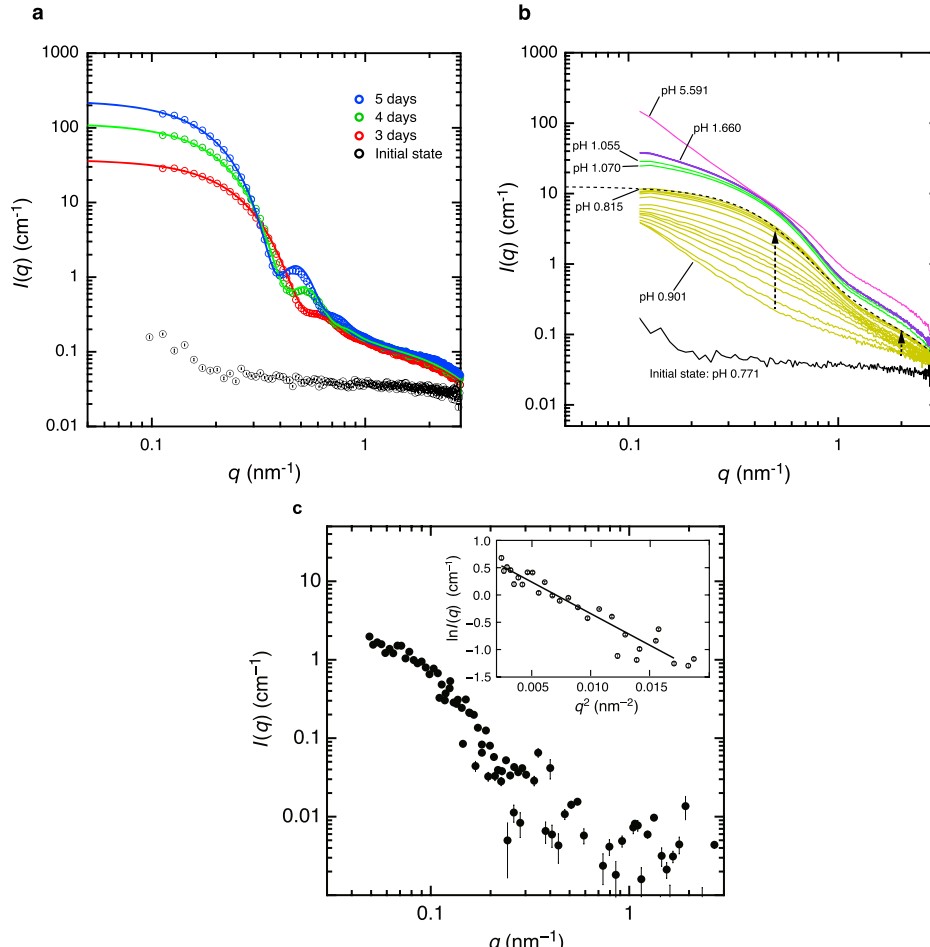

properties of the clusters to clarify which one is more appropriate for our system. The aggregation-fragmentation dynamics for active clusters was described by ref. 47. In their model, the cluster size distribution (denoted by $C_N$) starts from a binary reaction in the simplest formula:

$$\frac{C_N}{C_3} = \left[\prod_{m=4}^{N} \frac{C_1 A_{m-1}}{F_m}\right] = \frac{\kappa^{N-3}}{\sqrt{N(N-1)(N-2)/6}} \quad (4)$$

where $C_1$ is a normalisation constant and N is the number of primary particles. Finally, the cluster size distributions shown in Fig. 6a, b (pink dotted curves) satisfy Eq. (4) with $\kappa$ = 0.50/0.52[47]. The model treats N = 3 ($C_3$) as the minimum number for the particle aggregation.

Conversely, one assumes secondary cluster (B) aggregation using the simplest scheme for self-assembled micelles

$$A + A + A + \cdots + A = B \quad (5)$$

where A is the monomer of the primary cluster. The reaction equilibrium constant is $K = X_B/X_A^N$. Assuming $C = X_A + NX_B$ where $X_A$ and $X_B$ are the respective molar fractions, we obtain

$$X_A = \sqrt[N]{\frac{C - X_A}{NK}} \quad (6)$$

When $K \gg 1$ and $C \gg 1$, we have

$$X_A \leq \sqrt[N]{\frac{1}{NK}} =: f_N \quad (7)$$

In Figs. 6a, b, we compare the $f_N$ curves at $K = 10^0, 10^2, 10^4, 10^{80}$, and $10^{100}$. The critical aggregation concentration (CAC) reflects the transition between the primary and secondary clusters. Because this conventional model assumes $K \gg 1$, the values of $10^0$ and $10^2$ are not appropriate. This model does not consider secondary cluster fragmentation, thus simply yielding the speculated maxima curves for CAC. The primary and secondary clusters exhibit intrinsically different CACs.

Meanwhile, we calculate the secondary cluster concentration as $C_s$ = 6.97 mM (for details, see SI). The remaining $Ce^{IV}$ ($C_i$ = 43.0 mM) includes ionic species and primary clusters in the medium. Furthermore, the mean aggregation number of the secondary clusters is $\bar{n}$ = 27.0 from Fig. 6a (green circle). This is likely close to very low $K$, suggesting that the secondary clusters are loosely coupled together because the reaction is near equilibrium. The blue filled circle in Fig. 6b, where the two models (i.e., curves with pink dots and $10^{100}$) cross, yields $\bar{n} \sim 27$ (see Supplementary Table S1 and S2 and their relevant description in detailed calculation in Supplementary Note 1) and CAC ~ 0.20 mM. This CAC seems an underestimation. An exact solution must exist upwards from there: $K \sim 10^{57}$ and $K \sim 10^{26}$ are plausible equilibrium constants for the secondary and primary clusters, respectively, assuming that the observed $\bar{n}$ = 27.0 is the critical aggregation condition. Yet, this system is not completely elucidated without a precisely determined $K$, which is obtained by conversion from Gibbs free energy.

The phase transformation, in the other words, appears to be "Alder transition" in strong acid for the ballistic aggregation–dispersion model with a volume fraction $\phi$ of solid corresponds to 0.02. Course-grained primary clusters condense themselves as a globular secondary cluster with a highly-roughening surface (Fig. 7). The typical assumption often utilised for the electrical field between a charged point and flat plane is no longer applicable

**Table 1 | Summary of the parameters of hierarchical aggregates determined by analysis of the small-angle X-ray scattering intensity $I(q)$**

| t (days) | $R_p$ (nm) | $\sigma_p$ (nm) | $R_s$ (nm) | $\sigma_s$ (nm) |
|---|---|---|---|---|
| 3 | 0.82 | 0.22 | 9.24 | 1.29 |
| 4 | 0.83 | 0.22 | 11.17 | 1.31 |
| 5 | 0.84 | 0.21 | 12.41 | 1.29 |

$R_p$ radius of primary spherical particle, $R_s$ radius of secondary spherical particle, $\sigma_p$ and $\sigma_s$ are the respective standard deviations.

**Table 2 | Summary of particle diameters measured using different methods, and the half-width at half maximum (HWHM) used to calculate the particle diameter using Raman band widths and Eq. 1**

| Sample | Particle diameter (nm) | | | | HWHM (cm$^{-1}$) at 466 nm |
|---|---|---|---|---|---|
| | SAXS | TEM | Raman | Others | |
| CAN | — | — | | 2.59* | 25.0 |
| 1.00 M | Not measured | p: 1–3 | N. A. | | Not observed |
| 0.50 M | p: 1.64 | p: 1–3 | N. A. | | Not observed |
| 0.05 M | p: 1.68, s: 24.8 | p: 1–3, s: 20–30 | p: 0.98 | | 58.0 |
| CeO$_2$ | — | — | | 35.5** | 5.71 |

CAN ceric ammonium nitrate, SAXS small-angle X-ray scattering, TEM transmission electron microscopy.
*calculated from the empirical line, **measured by dynamic light scattering. p: primary cluster, s: secondary cluster.

in their closest separation. Instead, two spherical particles model is rather worthy in consideration: however, the roughening surface and asymmetrical 4–1 electrolyte are major difficulties for the present case. Because we have not proposed our theoretical model this time, the appropriate selection of known theory plays a vital role (see Theory).

Given the restriction with a repulsive interaction between electrical double layers, the classical DLVO theory predicts that the secondary cluster is stabilised near the local minimum in the static condition. However, many factors—e.g., rheological properties such as spatiotemporal heterogeneity of the strongly alkaline droplet, vigorous stirring, heat distribution, boundary condition—affect efficient energy transport in a multi-component complex fluid, giving rise to the Alder transformation which exhibits opalescence. Viscoelasticity intrinsically reflects a subtle change of surface potential due to strain stress among contact particles in the liquid. We recorded shear modulus ($G$ [dyn/cm$^2$]) as a function of volumetric fraction ($\phi$) to measure the potential among particles directly[48,49]. However, the result of the strain sweep curve was poor and not obvious enough to see a correlation between $\phi$ and $G$.

The approach considering diffusion limited aggregation (DLA) may be a more realistic pathway to transit via saddles or nodes on the free energy landscape of such a dynamic system. In fact, our in-situ real-time observation supports the idea of a complicated system with various parameters: the system falls into a limit cycle behaviour[50] under many conditions, and the only difference is when to participate. These factors induce the system of secondary clusters into a non-equilibrium state specified as *near equilibrium*[51]. A long lasting near equilibrium is visible in the SEM observation with physisorbed water remained (Fig. 8a–d). We recorded these images—clusters form agglomeration-flocculation—after 10 months of formation. Note that the mean diameter found here becomes larger three orders of magnitude than that found in Fig. 3a–c. The trend corresponds to the literature by dynamic light scattering.

Contrast-matching (hereafter, CM)-SANS profiles are vital to understand the role of counter-cations, i.e., NH$_4^+$ surrounding on the surface of ceria, on the hierarchical growth to the secondary aggregation of ceria. The

**Fig. 6 | The estimation of phase change as a function of concentration in various approaches.** Numerical curves of the critical aggregation concentration (CAC) for hierarchical nanoarchitectures as a function of the number of primary particles (N). Pink dotted curves are the cluster size distributions of secondary particles (converted using the analysis of ref. 47.). The curves are calculated using $CAC = f_N = (NK)^{-1/N}$. **a** shows the overall trend and (**b**) shows the expanded view. Blue "×": primary clusters, green square: secondary clusters. K is the equilibrium constant for secondary cluster formation. An arrow starts from the intersection to the curve with $K = 10^{57}$. **c** the observed turbidity [NTU] and estimated CAC as 54.6 mM. **d** The measured electrical conductivity [S/m] evaluates CAC as ca.53.0 mM.

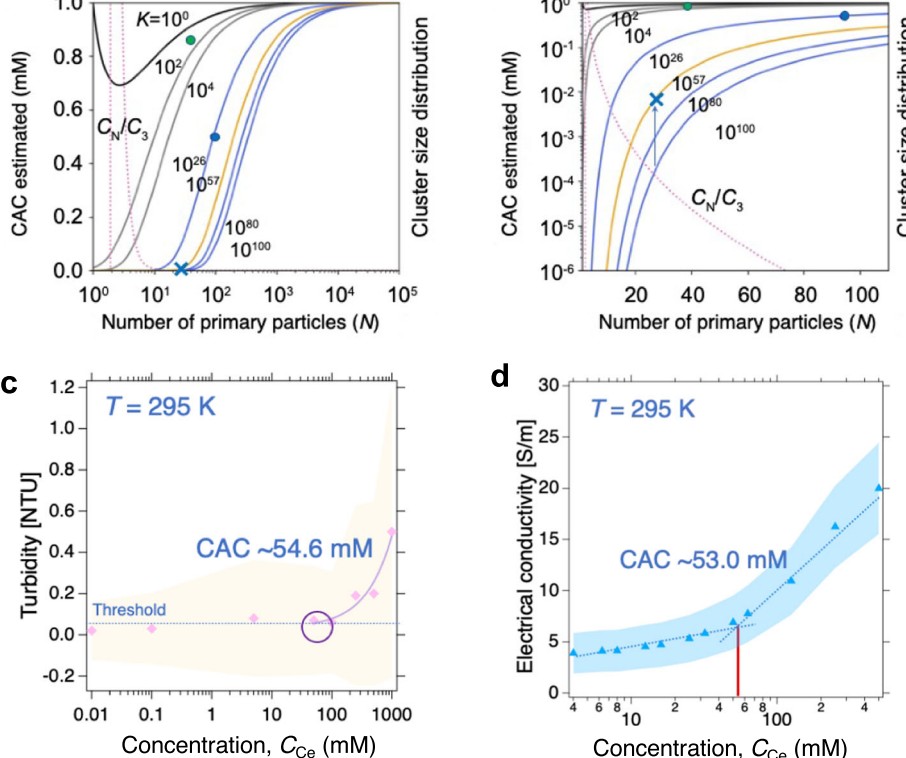

**Fig. 7 | Schematic drawings of surface interaction.**
**a** Depleted effect between two secondary clusters that are surrounded by primary cluster and less bulky complexes. **b** Roughly ordered surface parameters with averaged radius gyration $R_g \pm 3\sigma$. **c** Ammonium cation moiety on the external shear is highlighted by orange. **d** Non-planar roughening surface for a tetravalent ion that exists in the depleted area. **e–g** the comparison of double layers with $\kappa a$ in different ranges.

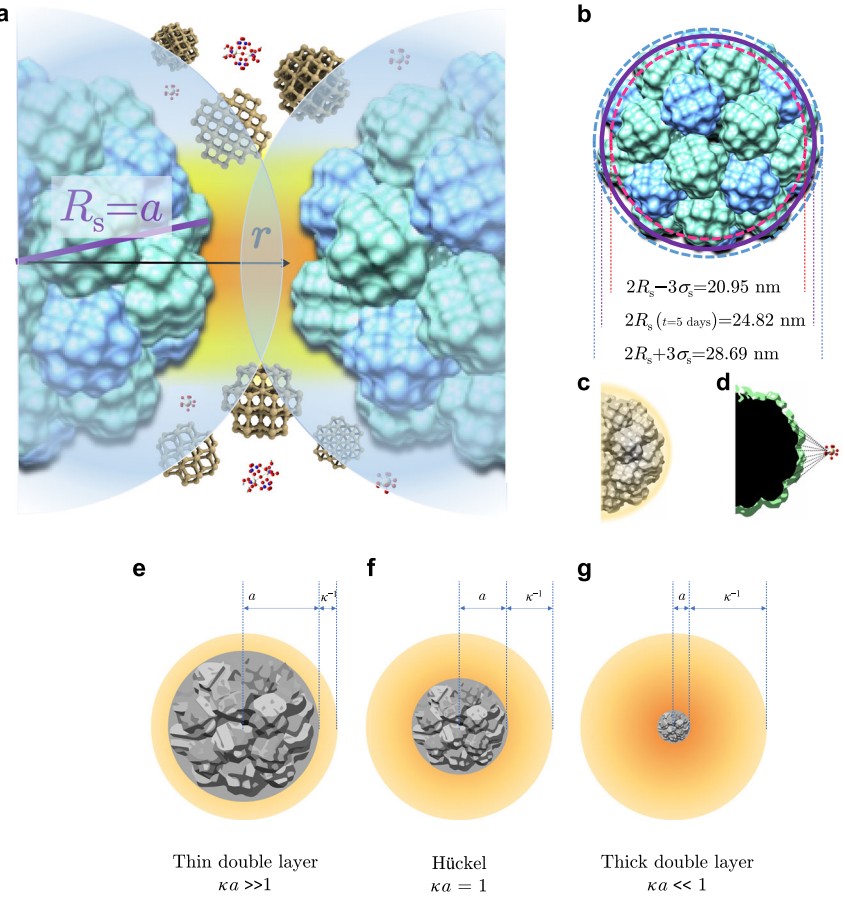

$2R_s - 3\sigma_s = 20.95$ nm

$2R_s \ (t=5 \ \text{days}) = 24.82$ nm

$2R_s + 3\sigma_s = 28.69$ nm

Thin double layer
$\kappa a \gg 1$

Hückel
$\kappa a = 1$

Thick double layer
$\kappa a \ll 1$

inner surface of ceria is positively charged at the very interface of ceria in the 0.1 M nitric acid medium, considering the point of charge (pzc) of ceria is *ca.* 8.1[52]. The possible counter anions within the Stern layer are mainly nitrate with a trace amount of hydroxide ions ($\sim 10^{-13}$ M) at the very local interface. Assuming these anions, the role of ammonium cations are more or less to generate the charge compensation between the shear plane and the interface of the diffuse double layer among primary clusters, bringing about the dispersive force to the large and highly charged ceria primary cluster—$Z_e \sim 10^n$ ($n > 2$). Because we failed to measure directly the total surface charge of the secondary clusters by zeta potential, we estimate the electrical conductivity $\sim 6.0$ S/m (CAC $\sim 53.0$ mM), from which is still difficult to convert a $Z_e$ value. The effort to determine the net bare charges are vital but seems rather ideal case so that an effective $Z_{eff}^*$ may play an alternative role[53]. Gillespie et al. studied counterion condensation essentially no free co-ions added. Shown in Fig. 5, we obtained the CM-SANS pattern of co-ion free system. Gyration radii, $R_{g,am}$ and $R_{s,Ce}$ ($t = 5$ days in Table 1.) estimated becomes the following order:

$$R_{g,am} = 18.7(\text{nm}) > R_{s,Ce} = 12.4(\text{nm}) \tag{8}$$

$R_{g,am}$ affords an averaged spherical distribution filled up for the secondary cluster of ceria moiety. The difference $\Delta r = R_{g,am} - R_{s,Ce} = 6.3$(nm) restricts $NH_4^+$ ions' maximum soft boundary. This evaluation is rather validated by considering a steep depleted region (Fig. 7a) to compensate charges among highly charged surface of the secondary cluster that are composed of tetravalent ions.

The first-order phase transition from the liquid state to solid state accompanying with dendrite pattern (Fig. 8i–l) by LCM-DIM is the evidence of existing homogeneous components in the system, whereas the heterogeneous complexed fluids with the primary cluster and the secondary

cluster (Fig. 8e–h) resulted in the glass transition, no systematic order, without such dendric crystal growth.

Besides the kinetic process, the fluctuation of water and water-related species is critical for fully describing the entire system involving local diffusion of primary and secondary particles by Brownian motion[54], potentially through a free hydroxide ion[55]. Furthermore, the dissipative structure is key to understanding mixed liquid-solid systems. However, these are just speculations at this time, and the validity of these theories should be checked in our subsequent work. The explanation for the hierarchical structure formation may be the surface free energy change under different conditions. This appears slightly paradoxical; and the present result represents a typical example within the Lyapunov stability[51] Self-assembly is ubiquitous in multi-component complex fluids, which paradoxically moderates the hierarchical reactions. Stability and instability are not only critical but complementary for co-optimisation in the nearby free energy landscape prior to bifurcation. Remarkably, Smarsly et al. recently reported $CeO_2$ colloidal particles similar to ours and proposed the synthetic pathways for particle formation[28]. However, their advanced work only characterised the agglomerates as products *via* their original protocols and did not perform in-situ observation of the secondary clusters. In this respect, the present study is not a simple extension of prior work, since it mainly focuses on the hierarchical self-assembly itself via in-situ observation for the determination of CAC and using CM-SANS and LCM-DIM observation. This approach enables us to cover the concentrations ranging from $10-3$ to 100 M, which are common conditions in condensed soft matter. However, it remains unclear how to explain the hierarchical nanoarchitecture in the multi-component mixture to reach an energetically relaxed state. subsequent studies on the boundary between equilibrium and nonequilibrium may have significant potential impact in nanoscience. Note that our model does not directly deal with a hierarchical structure. Rather, it only describes the relationship between two different scales. Scaling theory and

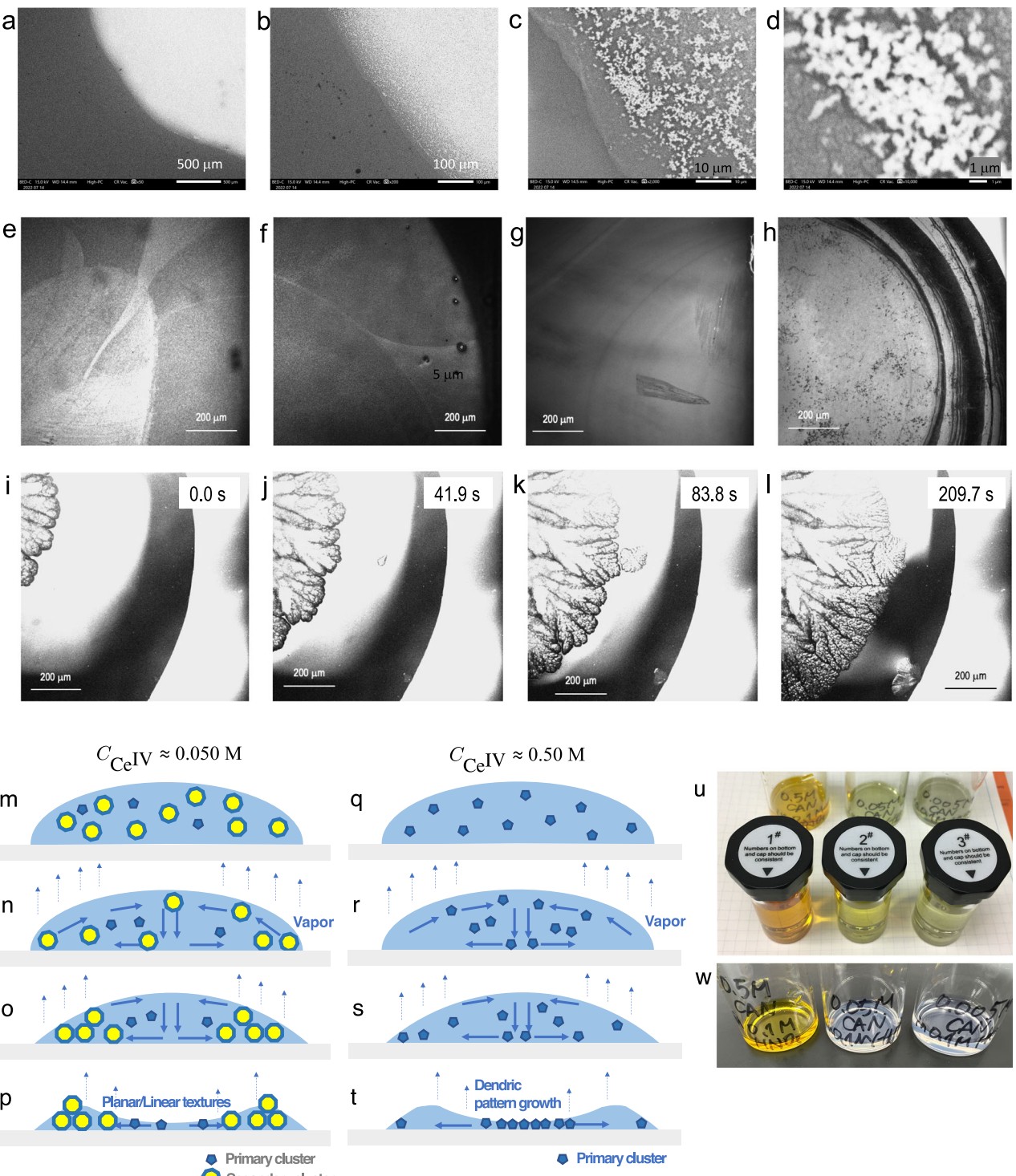

**Fig. 8 | Microscopic images of CeO₂-containing sub micro crystals on the quartz glass plate, schematic mechanism of opalescence, and samples. a–d** SEM images dried over two nights on the quartz glass plate without any coating pre-treatment. **a–h** The total concentration of added ceric ammonium nitrate is $C_{Ce^{IV}} \approx 0.050$ M in 0.10 M HNO₃. **e–h** Images taken by laser confocal microscopy combined with differential interference contrast microscopy (LCM-DIM). Different areas of interest in different droplets were observed. Various patterns forms: these images were taken at the air-acid interface under in-situ condition performed at $T_1 = 273$ K. **i–l** Dendric crystal growth patterns observed by LCM-DIM as time evolutes. Time starts from

10 min after settling at $T_2 = 323$ K. Scale: (**a**) 500 μm (**b**) 100 μm (**c**) 10 μm, (**d**) 1 μm, and (**e–l**) 200 μm. The total concentration of added ceric ammonium nitrate is $C_{Ce^{IV}} \approx 0.50$ M in 0.10 M HNO₃. **m–p** Scheme of flocculation of the secondary particles to result in structural colour (shown in **e–h**) as decreasing the volume of a droplet due to the addition of heat at 323 K. **q–t** Scheme of growth of dendric pattern as phase transformation from liquid to solid occurs due to the addition of heat at 323 K. **u–w** the emergence of weak structural colour (0.05 M, middle in (**u**); 0.005 M right in u) as time evolutes (**u** 0 day and **w** 90 days).

renormalisation group approaches developed in the field of condensed matter physics may render further insight. Additionally, it is theoretically possible to investigate the concomitant interaction within the hierarchical nanoarchitecture by calculating the force curves using state-of-the-art molecular simulations, which was not pursued in this study.

In summary, we conducted in-situ real-time observation of hierarchical $CeO_2$ nanoarchitectures formed from CAN in acidic solutions. Such observations of colloidal particles on a wide scale have rarely been reported. Scattering and microscopy were vital to verify the formation of a hierarchically assembled structure as a quasi-ceria colloid. The primary core clusters were 1–3 nm in size and composed of nanocrystals, while the secondary clusters were colloids 20–30 nm in size. Moreover, the secondary clusters exhibited colloidal aggregation and dispersion, which could not be speculated by the DLVO theory; nevertheless, it surely is a rather suitable case in the consideration of pair-G potential by the Sogami-Ise theory due to long range ordering giving rise to phase change: the Alder transition with a low volume fraction. The novelty in the present work is a reconsideration of the classical model in order to understand the dynamics by DLA approach, which explains the present results well. In particular, the surface attraction is enhanced between coarsely aggregated secondary particles is plausibly due to depletion-like or counterion condensation, leading to a higher order flocculation at the liquid surface and droplet edge. In such a multi-phase mixture, the complementary reactions of primary cluster formation at equilibrium and secondary cluster formation and hierarchical patterns above these molecular scales at near equilibrium represent a *live-and-let-live* principle to maintain a global efficient free energy network in the system (Supplementary Fig. S1). The dissipative structure and corresponding fluctuation may be critical in controlling this slow reaction. These results may provide a simple explanation for the geometrical structures of oligomeric complexes in solutions, although larger assembled structures in a complex fluid are more crucial for the development of Ce-based materials with opalescence or iridescence. This study using $CeO_2$ nanoparticles in a complex fluid is relevant to many other systems, and the results has substantial implications for colloid chemistry and non-equilibrium physics.

## Methods
### Samples
Commercially available CAN (purity >98%, Tokyo Chemical Industry Co., Ltd., Tokyo, Japan) was placed in an oven at 110 °C overnight to remove the physisorbed water. Concentrated $HNO_3$ (69 wt.%, Wako Pure Chemical Industries, Ltd., Osaka, Japan) and NaOH (50 wt.% solution, Wako Pure Chemical Industries, Ltd.) were diluted in milli-Q water for use as titrants during SAXS. For TEM, a droplet of CAN solution was placed on a copper microgrid (carbon-reinforced STEM 150Cu grid, Okenshoji Co., Ltd., Tokyo, Japan) and dried overnight at approximately 20 °C. For Raman microscopy analysis, pristine microcrystalline powders of dehydrated CAN and $CeO_2$ (Wako Pure Chemical Industries, Ltd., >99.9%) were mounted on a carbon seal. The $CeO_2$ particle size in 0.1 M $HNO_3$ was determined using a zetasizer (Zetasizer Nano, Malvern Panalytical, Malvern, UK) to be 83.4 nm for monomers and 354.7 nm for aggregates. Raman spectra were also collected for liquid samples placed on a glass plate. Conventional methods such as turbidity, electrical conductivity, and UV-Vis absorption spectrophotometry to evaluate the phase change of CAN in the liquid state were used. In the end, the contrast-matching small-angler neutron scattering (CM-SANS) was applied by adding tetraethylammonium nitrate in the $D_2O/H_2O$ mixture at the different volumetric fraction: (a) 100% $D_2O$, and (b) 66.1% $D_2O$.

### Procedure and apparatus
The formed hierarchical structures were examined in two CAN solutions with the total concentrations of $C_{Ce^{IV}}$ = 500 and 50 mM, denoted as the first and second series, respectively. TEM (JEM-2100F, JEOL Ltd., Akishima, Tokyo, Japan; operating at 200 kV)[56], In-situ Raman microscopy (Micro-Ram 300, Lambda Vision Inc.), and SAXS (Rigaku, Tokyo, Japan) analyses were used to obtain important information about the samples. The droplet

of solution samples was put on a glass plate. The micro-Raman spectrometer was equipped with a 532-nm Nd:YAG laser focused onto the sample using a $100 \times$ objective lens, and laser power at the sample position was 32 mW. The Raman spectra were recorded in the range 50–1970 $cm^{-1}$ with a spectral resolution of 4 $cm^{-1}$ and an acquisition time of 10 s. Each spectrum consisted of five accumulations acquired for each sample. Background subtraction was not performed. Raman spectroscopy was suitable at high concentrations of $C_{Ce^{IV}}$ = 0.50 and 1.00 M, where SAXS was no longer appropriate. Turbidity was determined using digital turbidimeter (TBD700, AS ONE Corporation). Electrical conductivity was measured by Benchtop Water Quality Metres (LAQUA F-74, HORIBA) hyphenated to conductivity electrode (3552-10D, HORIBA). UV-Vis spectrophotometer (V-550, JASCO) with thermostat jacket was used to record absorption spectra.

### SAXS
SAXS was performed using an X-ray diffraction apparatus (NANO-Viewer, Rigaku). The incident Cu Kα radiation ($\lambda$ = 0.154 nm) was focused on a spot 450 μm in diameter on the sample with a confocal optic (Max-Flux, Rigaku) equipped with a pinhole slit collimator. A two-dimensional (2D) position-sensitive detector (PILATUS 100 K/R, Rigaku) of $195 \times 487$ pixels with a spatial resolution of 0.172 mm/pixel was used to record X-rays scattered from the sample. The magnitude of the scattering vector, $q = (4\pi/\lambda)\sin\theta$, was between 0.1 and 2.8 $nm^{-1}$ in use of two sample-to-detector distances of 92 and 655 mm, where $2\theta$ is a scattering angle. We corrected the scattering data for the counting efficiency, instrument background, and air scattering on a pixel-to-pixel basis[57]. The X-ray scattering intensity distribution was circularly averaged and converted to an absolute unit of the scattering intensity ($cm^{-1}$) by calibration with water scattering[58]. The corrected scattering intensity was designated as $I(q)$, and then the cell scattering was subtracted from $I(q)$ by considering the transmission. The sample solution was loaded using a peristaltic pump into a titanium cell with a thickness of 1.25 mm (path length) and quartz plate windows (6.0 mm width × 2.5 mm height × 0.02 mm thickness)[57]. All X-ray scattering data were acquired at 25 °C.

**CM-SANS.** We performed CM-SANS measurements at SANS-J diffractometer, installed at research reactor JRR-3, Japan Atomic Energy Agency (JAEA), Tokai, Japan[59,60]. The wavelength of incident neutron beam was set to $\lambda$ = 0.65 nm with the wavelength distribution of $\Delta\lambda/\lambda$ = 0.15. The divergence of the incident neutron beam was defined by using a 20-mm-source aperture and an 15-mm-diameter sample aperture. Two two-dimensional position-sensitive detectors were used to detect scattered neutrons from the sample. The larger detector was $650 \times 680$ mm (length × width) and was composed of 95 $^3$He tube detectors 8 mm in diameter, and the smaller detector was $650 \times 385$ mm (length × width) and was composed of 48 $^3$He tube detectors 8 mm in diameter. Two sample-to-detector distances, $L$, of the larger detector were set as 4 and 2 m along the beam path of the incident neutrons, whereas that of the smaller detector was set in a high-angle region of the beam path of the incident neutrons with $L$ = 0.96 m. These configurations covered a $q$ region of 0.04 $nm^{-1} < q <$ 3 $nm^{-1}$. The two-dimensional scattering data were corrected on a pixel-to-pixel basis for counting efficiency, instrumental background, and air scattering. After circularly averaging the SANS intensity distribution, which is the one-dimensional scattering data; $I(q)$ versus $q$. the scattering intensity was converted to the absolute intensity unit of reciprocal centimetre ($cm^{-1}$) by using a secondary standard of an irradiated Al plate. The cell scattering was subtracted from the corrected scattered intensity by considering the sample transmission.

We employed the contrast matching technique using $D_2O/H_2O$ (= 0.6618/0.3382, v/v) mixture as a solvent matrix in CM-SANS experiment. The SLD of the $D_2O/H_2O$ mixture, $\rho D_2O/H_2O$ = $4.02 \times 10^{-10}$ ($cm^{-2}$), agrees with the that of $CeO_2$. This means the $D_2O/H_2O$ and $CeO_2$ can be regarded as same medium in the CM-SANS measurements, and hence, the scattering contribution from the ammonium ions only is capable of observing during the hydrolysis reaction process of CAN. It is worth noting that SAXS focuses

on the building process of $CeO_2$ secondary particles consisting of the primary particles, whereas CM-SANS enables us to observe the ammonium ions. This kind of complementary approach using both SAXS and CM-SANS provides insights into mechanism of $CeO_2$ hierarchical colloidal formation. The reaction mixture was loaded into quart cell with 2 mm thick. The quartz cell was placed in the incident beam path of neutrons at 25 °C. The measurement time of 600 s or 1200 s were spent for acquiring CM-SANS data at $L = 2m$ and $4m$, respectively. Sample preparation was identical to the condition (see caption in Fig. 5c). Figure 5c shows the CM-SANS profile obtained for the solution with $C_{Ce^{IV}} = 50.0$ mM in 1.0 M $HNO_3$ using a $D_2O/H_2O$ mixture (= 0.6618/0.3382, v/v), which enables us to detect the scattering contribution of the ammonium cations only due to the contrast matching condition with $CeO_2$ particles. Namely, in this situation, the scattering length density (SLD) of the $D_2O/H_2O$ mixture agrees with that of $CeO_2$. The small-angle scattering intensity at $q < 0.4$ nm$^{-1}$ increased with decreasing $q$, while that at $q > 0.4$ nm$^{-1}$ was relatively weak and close to the noise level of the instrument. We speculate that the small-angle scattering at $q < 0.4$ nm$^{-1}$ originates from the inhomogeneous distribution of ammonium cations in solution. Assuming the aggregates of the ammonium cations, the radius of gyration, $R_{g,am}$, was evaluated based on the Guinier approach (c.f. Supplementary Fig. S2). The inset of Fig. 5c shows the Guinier plots, which give a linear relationship between $\ln I(q)$ and $q^2$ for $qR_{g,am} < 1$. The radius $R_{g,am}$ was then determined from the slope, corresponding to $-\frac{1}{3}R_{g,am}^2$, and thus, resulting in $R_{g,am} = 18.7$ nm.

**Laser confocal microscopy (LCM) combined with differential interference contrast microscopy (DIM)[61–63].** The further detailed description of the apparatus is found elsewhere. Here, we show a brief introduction. LCM can significantly reduce noise. DIM is a traditional optical microscopy technique that gives a three-dimensional-like contrast. Then, the combination of LCM and DIM (LCM-DIM) can visualise individual molecular layers (0.4 nm in thickness) on ice crystal surfaces. We used a confocal system (FV300, Olympus Optical Co. Ltd.) attached to an inverted optical microscope (IX70, Olympus Optical Co. Ltd.). A super luminescent diode (ASLD68-050-B-FA, 680 nm, Amonics Ltd.) was used for LCM-DIM observations. A droplet of 5.0–50.0 μL containing sample liquid was on a home-crafted quartz glass cell that was placed on the home crafted heating/cooling stage at a range of 278–323 K.

Macro-aggregating behaviour often terms as agglomeration-flocculation. As it turns out that these structures based on nanoarchitectures form patterns beyond micro-metre planar scales, we employed laser confocal microscopy combined with differential interference contrast microscopy (LCM-DIM). This advanced optical microscopy enables us to observe wide range (sub-millimetres) phase transformation; moreover, the phase transition dynamics along z-axis (perpendicular to the samples' surface) is observable as a function of temperature.

**Rheology.** We recorded the viscoelastic property of acidic colloidal suspensions at room temperature ($T_R = 293$ K) (Anton Paar, SUS304 ϕ50 mm parallel plate hyphenated to Rheometer Dynamic Mechanical Analyzer MCR702). In a series of samples, a pipetted amount of 50 μL was let on a stainless stage. Complex shear moduli ($G^* = G' + iG''$) was recorded with the angler frequency hold at 10.0 (rad/s), where slight plateau is provided. We compare the trend in a series of $G^*$(dyn/cm$^2$) where strain sweep (γ in %) value becomes *ca.* 1.00.

**Titration trends**
We observed the colloidal behaviour of the system near equilibrium[64], including nanocrystal aggregation and the growth of these nanocrystals upon addition of a strong alkaline solution to alter the pH. Figure 5b shows SAXS patterns obtained during the titration of CAN ($C_{Ce^{IV}} = 50$ mM) in 1.0 M $HNO_3$ using 19.7 M NaOH. During titration, the pH increases from 0.77 to 5.59. Prior to NaOH addition (pH 0.77), the weak scattering intensity (black line) indicates no ordered structure, and this SAXS pattern is identical to the initial state in Fig. 5a. Upon adding the first droplet of concentrated

NaOH (2 μL) to 1.0 mL of CAN solution, the pH increases to 0.901, with an immediate, severe change in the SAXS pattern (yellow lines). The SAXS intensity increases with decreasing $q$, resulting in the power-law scattering of $I(q) \sim q^{-2}$ at $q < 0.5$ nm$^{-1}$, which indicates the formation of large aggregates that may be linked to promoted hydrolysis at higher local pH. After another 3.5 h, the pH gradually decreases to 0.82, and the SAXS intensities at approximately $q = 0.5$ and 2.0 nm$^{-1}$ gradually increase to form two shoulders (indicated by dashed arrows). This suggests a hierarchical ordering within the solute, specifically the primary clusters and secondary aggregates.

The SAXS pattern at pH 0.82 (top yellow line) is reproduced using Eq. (2) with the refined parameters $R_p = 0.85$ nm, $R_s = 5.31$ nm, and $\sigma_R = 1.51$ nm, as indicated by the best-fit theoretical profile (black dashed line). $R_s$ becomes less than half of that in Fig. 5a, whereas $R_p$ shows good agreement. Remarkably, in the SAXS patterns the first peak of $I_s(q)$ was not observed due to a broad distribution ($\frac{\sigma_R}{R_s} = 0.28$), suggesting that the primary cluster configuration within the secondary aggregates likely persists, even in a metastable state.

The second droplet of NaOH (6 μL) induces an abrupt increase in pH from 0.90 to 1.07. Accordingly, the SAXS intensity increases without significant changes in the $q$ dependence (lower green line). The SAXS pattern does not change without further addition of NaOH, though the solution pH decreases to 1.06 after 12 h (top green line). The third droplet of NaOH (4.5 μL) adjusts the pH to 1.66 and induces a slight increase in the scattering intensity at $q < 0.4$ nm$^{-1}$ while maintaining that at $q > 0.4$ nm$^{-1}$ (purple line). This indicates further aggregation among the secondary aggregates. The pH changes very little with time (barely on the order of 0.01) for each short SAXS measurement. After adding a fourth NaOH droplet (1.5 μL, pH 5.59), the scattering intensity at $q < 0.4$ nm$^{-1}$ increases dramatically (pink line) while the shoulder peak at $q \approx 2.0$ nm$^{-1}$ remains, suggesting primary clusters within the system. At this high pH, the solution becomes turbid due to $[Ce(OH)_4]_n$ formation, finally forming a muddy precipitate.

**Computer software**
Molecular modelling was carried out using UCSF Chimera package (www.cgl.ucsf.edu/chimera/)[65]. The CAC curves were produced using matplotlib and seaborn v0.8.1[66]. Some of the images and videos were generated using CrystalMaker® 10.5.4 (www.crystalmaker.com). The SAXS patterns were analysed using Igor Pro (Wavemetrics, Lake Oswego, OR, USA).

**Reaction modelling**
For the hydroxo-bridging ligands such as $\mu$-OH and $\eta$-OH, SAXS revealed a hierarchical structure with an aggregated shape of rigid globules, because the primary and secondary clusters grew at different rates. To the best of or knowledge, such results have not been reported in the literature except for Smarsly et al.[28] possibly due to a lack of advanced techniques or interest. Ions, complexes, dimers, trimers, 38- and 40-mers, *etc*[21]. and their aggregates co-exist in a single solution. Therefore, we extend Eq. (3) as follows:

$$m\mathrm{Ce}^{4+} + n\mathrm{OH}^- \underset{k_{-m}}{\overset{k_m}{\rightleftharpoons}} \sum_{\substack{0 \leq i \leq m \\ 0 < j < n}} \mathrm{Ce}_i(\mathrm{OH})_j^{(4i-j)+} \tag{9}$$

with

$$\sum_{\substack{0 \leq i \leq m \\ 0 < j < n}} \mathrm{Ce}_i(\mathrm{OH})_j^{(4i-j)+} := \{\mathrm{Ce}_1\} + \{\mathrm{Ce}_2\} + \{\mathrm{Ce}_3\} + \cdots$$
$$+ \{\mathrm{Ce}_i\} + \cdots + \{\mathrm{Ce}_{38}\} + \cdots + \{\mathrm{Ce}_m\} \tag{10}$$

where the kinetic constants $k_m \geq k_{-m}$, and $\{\mathrm{Ce}_i\}$ denotes the $i$-mer of hydroxo-species. Solid-state $CeO_2$ is formed in the limit of $m \rightarrow \infty$, yielding the irreversible kinetic constants $\mathrm{k}_m \gg \mathrm{k}_{-m}$.

The assumption $a_i = \{Ce_i\}$ holds when all species exist in equimolar fractions. Considering the cluster-cluster dimer, we approximate Eq. (10) by the *self-assembly reaction*:

$$\sum_{\substack{0 \le i \le m \\ 0 < j < n}} Ce_i(OH)_j^{(4i-j)+} \sim \left(\sum_i a_i\right) - \mathcal{O}\left(g\left(\binom{m}{k}^k\right)\right) \qquad (11)$$

where $\mathcal{O}(g(X^k))$ is the Bachman-Landau $O$-notation of the function $g(x)$, and $X^k = \binom{m}{k}^k$ is the $k$th order of a binomial distribution. The conventional solution chemistry approximates the equilibria in a dilute solution. The first term in the summation on the right side of Eq. (11) considers the formation of multinuclear species with equivalent or different numbers of OH⁻. The second term with the $O$-notation represents the order of reaction, including the probability of collision between two primary clusters. The second term in Eq. (11) becomes non-negligible when interparticle interaction is dominant, resulting in the aggregation of k particles out of a total of m particles. Here, the solution is close to a regular one, and thus the second term on the right is no longer negligible and potentially becomes dominant. Thus, we adopt a modelling approach suitable for aggregation–fragmentation dynamics, and compare this model with the classical model that approximates the system's conditions. Taken together, these models should cover the two-rank hierarchical system.

Meanwhile, the SAXS data during titration contain a counterintuitive feature: when more base is added, the pH decreases instead of increases. Assuming $n = 4\,m$ in Eq. (11), we consider the formation of the solid phase:

$$m Ce^{4+} + 4m OH^- \rightleftharpoons m[Ce(OH)_4](aq) \qquad (12)$$

$$[Ce(OH)_4](aq) \underset{k_{-1}}{\overset{k_1}{\rightleftharpoons}} [Ce(OH)_4](s) 0+ \qquad (13)$$

In our system, the kinetic constant $k_1$ in Eq. (13) is too large to be experimentally measurable. Instead, the following metal ion polymerisation presumably emerges as primary cluster nucleation:

$$m[Ce(OH)_4](aq) \underset{k_{-m}}{\overset{k_m}{\rightleftharpoons}} m[Ce(OH)_4](s) \qquad (14)$$

Equation (14) explains non-equilibrium in the classical thermodynamic description for a sufficiently large m. However, it could not possibly occur on the atomic scale. The SAXS patterns in the high-$q$ range ($q > 1.0$) remain almost intact or show a little reduction in terms of $I(q)$ as a function of t (data represented by open circles in Fig. 5a). This supports the literature descriptions regarding nanocrystalline CeO₂ solubility [67].

**Theory.** Based upon the Debye-Hückel theory, the Sogami-Ise theory to our study provides an improved basis. The Debye-Hückel approximation is strictly applicable only in the case of low potentials. With tetravalent and asymmetrical ions highly present in our system, the theory is no longer straightforward due to the restriction. The application of the Sogami-Ise theory finds much more valid than the classical DLVO theory because the latter is not guaranteed for the system with low concentration that undergoes the phase transformation due to long range interaction such as strong attractive force. The Sogami-Ise theory insists an extensive occasion with phase changes under a variety of salt concentration. However, the present experimental result of the shear moduli was not qualitatively satisfactory for the comparison of the theoretical approaches. This fact is presumably due to the surface roughening of our particles and condition which is the salt free limit. Therefore, we limit our discussion to the frequently-argued model—counterion condensation. Whereas the ceria surface in 0.1 M nitric acid has a high charge, there is no direct observation this time. Highly charged surface in a salt free system has less attention except few cases[53]. The 4–1 electrolyte pair has

very little examples in the Poisson-Boltzmann equation[68]. Also, the Debye screening length for CeⅣ has not been reported[69,70]. We simply applied salt concentration as 0.1 M ($C_{Ce} = 0.05$ M); however, this is no longer appropriate for theestimation of $\kappa^{-1} \sim 76.6$ nm due to the too small scale of the primary clusters[48]. And thus, we obtain $\kappa a \sim 0.3 < 1.0$. This is the case where double layer thickness is comparable to the radius gyration of the secondary particles. As was put forward by Bartlett et al., Manning's counterion condensation is rather more suitable for consideration in the present system [53].

## Data availability
The authors confirm that the data supporting the findings of this study are available within the article and its supplementary material.

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

## Acknowledgements

This work was supported by the Japan Society for the Promotion of Science (JSPS) KAKENHI (Grant Number JP20K05387 and 23K17808) and Ministry of Education, Culture, Sports, Science and Technology (MEXT) KAKENHI (Grant Numbers JP18KK0148, JP18H01921, and 22H02010). N.A. is grateful to the Advanced Science Research Centre (ASRC), Japan Atomic Energy Agency (JAEA) for the REIMEI Research Program 2021–22 and Houga Research (2023–24). Access to the experiment at the SANS-J (C3-2) was provided by the Japan Research Reactor-3 (JRR-3) with the approval of Japan Atomic Energy Agency (JAEA; Proposals 2021-D400, 2022-D586, and 2023-D857.)

## Author contributions

N.A. directed the original project. N.A., R.M., and A.I.-O. guided the research. All authors interpreted the results and contributed to the manuscript. N.A., S.N., and R.M. performed SAXS studies and R.M. analysed the data. R.M. and Y.U. performed CM-SANS measurement. T.T. performed TEM. T.S. conducted zetasizer studies. T.Y. performed Raman microscopy. N.A. prepared the samples. N.A. and M.O. performed theoretical modelling. G.S. performed LCM-DIM observation.

## Competing interests

The authors declare no competing interests.

## Additional information

**Peer review information** *Communications Chemistry* thanks the anonymous reviewers for their contribution to the peer review of this work. This manuscript has been previously reviewed at another Nature Portfolio journal. This document only contains reviewer comments and rebuttal letters for versions considered at Communications Chemistry. A peer review file is available.

