## [Peer Review File · Communications Chemistry]

This manuscript has been previously reviewed at another Nature Portfolio journal. This document only contains reviewer comments and rebuttal letters for versions considered at *Communications Chemistry*.

REVIEWERS' COMMENTS:

Reviewer #3 (Remarks to the Author):

I have reviewed this manuscript for several rounds. The authors have been very serious about the comments and addressed all issues in very details. Now the manuscript can be accepted.

Reviewer #5 (Remarks to the Author):

The task I was asked to is to assess if the authors have properly responded to the comments by reviewer 1, and in short, they do. In fact, the authors have gone to great lengths to modify the manuscript in both the first and second revision. I'm further impressed by the multi-technique approach (including theory) that the authors have employed.

I have concerns that are similar to those put forward by the original reviewers: (1) the language is often unclear, e.g. "...it is an incomparable approach to make an in-depth understanding the patterns formed by..." and (2) the real significant step with respect to ref. 27. In my opinion the manuscript does not quite reach the standards expected in Communications Chemistry, but the view of the other reviewers should be dominant in determining the final verdict.

Response to the Reviewer #5's comments

The point by point response to Reviewer comments

Reviewer #5

□ Reviewer #5 Overall Comments 1.0

REVIEWERS' COMMENTS:

Reviewer #5 (Remarks to the Author):

The task I was asked to is to assess if the authors have properly responded to the comments by reviewer 1, and in short, they do. In fact, the authors have gone to great lengths to modify the manuscript in both the first and second revision. I'm further impressed by the multi-technique approach (including theory) that the authors have employed.

I have concerns that are similar to those put forward by the original reviewers: (1) the language is often unclear, e.g. "...it is an incomparable approach to make an in-depth understanding the patterns formed by..." and (2) the real significant step with respect to ref. 27. In my opinion the manuscript does not quite reach the standards expected in Communications Chemistry, but the view of the other reviewers should be dominant in determining the final verdict.

Response 5.1

First, we greatly appreciate the valuable effort and insightful comments from reviewer #5. We realise that the description in the second edition (page 5, line 110) was not easy to understand. Thus, in the third revision we rephrased it as follows:

~~Because the finding in controlling the alignment of aggregated particles based on nano scaled modelling for globular colloid with tetravalent metal ion is lacking, it is an incomparable approach to make an in-depth understanding the patterns formed by these materials that potentially exhibit photo-physical properties—opalescence or iridescence.~~

The lack of knowledge on how to control the alignment of aggregated particles based on nanoscale modelling for globular colloid with tetravalent metal ion is an unparalleled approach to gain an in-depth understanding of the patterns formed by these materials, which may exhibit photophysical properties - opalescence or iridescence.

The real significant step with respect to the corresponding reference was argued in the response 2.2 in the second response letters. The new significant finding is that our experimental comparison of SAXS with that of SANS focuses a spot on the surface interaction between the secondary particles with the depletion-like or counterion condensation between the rough surface of them. This was not accessible without elucidating surface interaction at molecular scale. However, we have not concluded the osmotic pressure with small cations, this is a subject in the following work. The advantage in understanding this is to exploit the self-assembly properties in the aligned packing of these photonic particles. LCM-DIM demonstrates that particle alignment is possible by preparing an optimal droplet to dry out the packing of particle domains. All these aspects are not found in the corresponding reference. In summary section, we added the following sentences (colored in purple with yellow):

The novelty in the present work is a reconsideration of the classical model in order to understand the dynamics by DLA approach, which explains the present results well. In particular, the surface attraction is enhanced between coarsely aggregated secondary particles is plausibly due to depletion-like or counterion condensation, leading to a higher order flocculation at the liquid surface and droplet edge. In such a multi-phase mixture, the complementary reactions of primary cluster formation at equilibrium and secondary cluster formation and hierarchical patterns above these molecular scales at near equilibrium represent a *live-and-let-live* principle to maintain a global efficient free energy network in the system. The dissipative structure and corresponding fluctuation may be critical in controlling this slow reaction.

(The following is cited from the second response letter)

Response 2.2

Comment 2.2 of reviewer #2 and comment 4.8 of reviewer #4 correspond. They both claimed that the original manuscript was a reiteration work of what Smarsly et al. found before our work.

Smarsly's paper was published while we were preparing and submitting our paper. We were in no way influenced by Smarsly's paper in developing our paper, and we were stunned to find that the content was very similar. However, it is an undeniable fact that Smarsly's paper was published first and we take this fact seriously. The revised paper is different from the first submitted paper and can be clearly distinguished from Smarsly's paper.

The specific differences are 1) our approach and newly applied methods to comprehend the order number of hierarchies, 2) the detailed building block picture to connect clusters' phase between hierarchies through *in-situ* observation using Raman spectroscopy, LCM-DIM, SAXS, and SANS, and 3) the understanding of surface interaction resulting in phase change appearing as a larger scale than secondary aggregates. A few more details are as follows:

- 1) We have revised our manuscript to emphasize the *in-situ* analyses in detail, focusing more on the hierarchical structures and patterns. We also add and rewrite new comprehensive methods—SANS, LCM-DIM and physical properties such as electrical conductivity and turbidity. This approach is to spot more light on the surface interaction in the liquid state. With the integrated understanding from these results, we achieve a more comprehensive detailed picture on at least three different levels of hierarchies.
- 2) The scheme below this sentence is the known clusters of Ce-containing materials. We have tried to crystalize smaller complexes to clearly state the individual structure and bonding on these matters. On the other hand, the present manuscript aims at considering one of these complexes as primary clusters so as to describe the second order aggregation and dispersion nature among formed complexes. Of course, we have to put away with the little story from our opinion in the refined article, which was improved by the comment of reviewer #4. Here, we have newly added the electrical conductivity and turbidity to quantitatively evaluate the critical aggregation concentration (CAC); furthermore, we have newly added SANS patterns and laser confocal microscopy combined with differential interference microscopy (LCM-DIM) to observe *in-situ* hierarchical ceria nanoarchitectures.

Fig. R1. Isolated Ce(IV) complexes* in the literature. (*{Ce₃} is from a theoretical calculation)

- 3) What we found out in the revised manuscript is a unique behavior of the extensive secondary particles' flocculation of secondary particles to obtain textile patterns as shown in Fig.8 e–h. The liquid suspension exhibits structural colors—a common photo-optical property known as iridescence. Using these techniques opens up a completely innovative direction to gain an in-depth understanding of the in-situ characteristics of the aggregation-flocculation of the secondary particles. In fact, the revised manuscript has much emphasis on the formation of new photophysical properties. The overall surface interaction including ammonium ions, confirmed by SANS, gave rise to these properties.

We have made a significant correction on the last part of the Introduction as follows:

“However, reports of atomically precise CeO₂ nanocrystal synthesis remain limited.¹ Although researchers continue to fill gaps in the cluster size from monomer, dimer, trimer, and hexamer to {Ce₂₄^{IV}}, {Ce₃₈^{IV}}, {Ce₄₀^{IV}}, and very recently {Ce₁₀₀^{IV}}; that is, the results remain incomplete. Another future research topic is controlling the aggregation of Ce^{IV} nanoclusters. CAN possesses two moles of anion relative to one mole of cation, presenting a typical multiscale system with charged ions or complexes, nanocrystal growth, and interactions among colloidal particles (Fig. 1). Remarkably, Smarsly et al. reported a first observation of hierarchical growth and aggregation of ceria colloid. Regardless of these successful efforts to obtain clusters, there has been no other hierarchical patterns except globular aggregation. Because the finding in controlling alignment of aggregated particle is lacking, it is a vital approach to precisely cast these materials that potentially exhibit photo-physical properties—iridescence. This study aims to observe in real time the crystal seed formation, aggregation and further extensive flocculation toward CeO₂ nanoparticles and subsequent *in-situ* patterns in a CAN-HNO₃ solution and carry out the corresponding modelling. The system involves two/three spatiotemporally different patterns that commonly emerge in reaction–diffusion processes of condensed complexed fluids with multiple components. Thus, it is relevant to a broad range of phenomena in the design and synthesis of metal oxides with precisely controlled size, structure and texture. Furthermore, the resulting nanostructured CeO₂ has potential applications in catalysis and other areas wherein the interface of nanoarchitectures is critical.”

We have also added new figures 5c, 6c, 6d, 7 and 8. The same applies to the corresponding sentences:

References

- 1 Mitchell, K. J., Abboud, K. A. & Christou, G. Atomically-precise colloidal nanoparticles of cerium dioxide. *Nat. Commun.* **8**, 1445 (2017). <https://doi.org/10.1038/s41467-017-01672-4>